# Predictive Value of GINI and ALBI Grades in Esophageal Cancer Receiving Chemoradiotherapy

**DOI:** 10.3390/curroncol31110504

**Published:** 2024-11-02

**Authors:** Timur Koca, Busra Hasdemir, Rahmi Atıl Aksoy, Aylin Fidan Korcum

**Affiliations:** 1Department of Radiation Oncology, Akdeniz University School of Medicine, Antalya 07070, Turkey; bhasdemir@akdeniz.edu.tr (B.H.); aylinkorcum@akdeniz.edu.tr (A.F.K.); 2Department of Radiation Oncology, Izmir City Hospital, İzmir 35510, Turkey; rahmiatil.aksoy@saglik.gov.tr

**Keywords:** esophageal cancer, albumin–bilirubin grade, global immune–nutrition–inflammation index, biomarkers, radiotherapy, chemotherapy

## Abstract

**Objectives:** The principal objective of this study was to assess the predictive efficacy of the global immune–nutrition–inflammation index (GINI) and the albumin–bilirubin (ALBI) score among patients receiving chemoradiotherapy for esophageal cancer. **Methods:** A retrospective analysis was conducted on 46 patients who received definitive or neoadjuvant radiotherapy for esophageal cancer at our institution. Blood samples were collected from these patients prior to the initiation of radiotherapy to measure the biomarkers, including the C-reactive protein (CRP), neutrophil–lymphocyte ratio (NLR), platelet–lymphocyte ratio (PLR), monocyte–lymphocyte ratio (MLR), the global immune–nutrition–inflammation index (GINI), and the albumin–bilirubin (ALBI) grade. The predictive significance of these biomarkers for progression-free survival (PFS) and overall survival (OS) was evaluated using both univariate and multivariate Cox regression analyses. **Results:** The median follow-up time for this study was 19.5 months (range: 2.6–166.3 months). Univariate analysis revealed that the platelet count (*p* = 0.003) and monocyte count (*p* = 0.04) were significant predictors of PFS. In the multivariate analysis, only the platelet count (*p* = 0.005) remained an independent predictor of PFS. Univariate analysis demonstrated that the neutrophil count (*p* = 0.04), lymphocyte count (*p* = 0.01), NLR (*p* = 0.005), PLR (*p* = 0.004), CRP (*p* = 0.02), ALBI grade (*p* = 0.01), and GINI (*p* = 0.005) were significant predictors of OS. Multivariate analysis identified the GINI as a predictor of OS, approaching statistical significance (*p* = 0.08). **Conclusion:** The results of our study indicate that the pretreatment GINI and ALBI grades are significantly and independently associated with the OS rates in patients with esophageal cancer who are undergoing chemoradiotherapy.

## 1. Introduction

Esophageal cancer is the eighth most diagnosed malignancy and ranks as the sixth leading cause of cancer-related mortality globally [1]. The variations in its prevalence among different countries and populations can be attributed to differences in the risk factor prevalence and the distribution of various subtypes. This malignancy is primarily classified into two histological subtypes: squamous cell carcinoma (SCC) and adenocarcinoma (AC), each associated with distinct sets of recognized risk factors. SCC is associated with alcohol consumption and smoking, while AC is commonly linked to Barrett’s esophagus, a history of gastroesophageal reflux disease, tobacco use, and obesity [2]. For cases of locally advanced esophageal cancer, the treatment options include neoadjuvant chemoradiotherapy followed by surgical intervention or definitive chemoradiotherapy [3].

The albumin–bilirubin (ALBI) grade is a novel index developed to assess the liver function and predict the prognosis in patients with hepatocellular carcinoma (HCC), independent of the degree of underlying liver fibrosis. The prognostic significance of the ALBI grade has been extensively documented across various treatment modalities for HCC, including hepatic resection, systemic therapies, locoregional ablative therapies, and transarterial treatments [4]. Furthermore, the ALBI grade has demonstrated prognostic utility in a range of other malignancies, including pancreatic, colorectal, and gastric cancers, as well as intrahepatic cholangiocarcinoma, extrahepatic bile duct cancer, and certain brain tumors, such as gliomas and medulloblastomas [5,6,7,8,9,10].

An expanding body of evidence indicates that chronic inflammation, recognized as the seventh hallmark of cancer, is integral to nearly every stage of carcinogenesis and disease progression, encompassing processes from uncontrolled cellular proliferation to the development of metastasis [11]. The inflammatory response typically encompasses neutrophils, monocytes, platelets, and lymphocytes, as well as the cytokines and chemokines these cells secrete, alongside acute-phase reactant proteins such as albumin and C-reactive protein (CRP) produced by various cells in the body. Research into the prognostic and predictive value of these inflammatory biomarkers in various cancers is increasing, and new biomarkers are continually being developed. The global immune–nutrition–inflammation index (GINI), which integrates multiple cellular and biochemical markers of inflammation, has been studied in patients with non-small-cell lung cancer (NSCLC). The findings indicate that the GINI serves as a statistically significant predictor of survival outcomes in NSCLC patients [12]. However, the predictive value of the GINI and ALBI grades in patients with esophageal cancer who are undergoing chemoradiotherapy has not yet been evaluated. Therefore, the principal objective of this retrospective study was to assess the predictive efficacy of the GINI and ALBI grades among patients undergoing definitive or neoadjuvant therapy for esophageal cancer.

## 2. Materials and Methods

### 2.1. Patient Selection

Approval for this retrospective study was obtained from the Clinical Research Ethics Committee of the University Medical Faculty. This study analyzed data from patients with esophageal carcinoma who underwent definitive or neoadjuvant treatment at the University Medical Faculty Hospital between January 2013 and January 2024. The exclusion criteria included patients under 18 years of age, those with an Eastern Cooperative Oncology Group Performance Status (ECOG PS) greater than 2, those presenting with distant metastases, and patients treated with palliative intent. In addition, patients who had chronically active immune or inflammatory diseases and active infections were excluded. This retrospective study followed the relevant REMARK guidelines [13]. All the patient data were deidentified.

### 2.2. Treatment Details

The treatment protocol included concurrent chemotherapy (carboplatin and paclitaxel, cisplatin alone, cisplatin and 5-FU, or others) and radiotherapy. Induction chemotherapy or adjuvant chemotherapy was applied according to the decision of the medical oncologist.

Computed tomography (CT) simulations were performed, with a slice thickness of 2.5 mm, for radiotherapy portal contouring. For better delineation of the gross tumor volume (GTV), positron emission tomography-CT (PET-CT) and/or magnetic resonance imaging (MRI) was fused with the planning CT at the time of initial diagnosis. The GTV encompassed the primary tumor along with any affected regional lymph nodes. The clinical target volume (CTV) was defined as the primary tumor plus a cranio-caudal expansion of 3 to 4 cm along the length of the esophagus and cardia, combined with a 1 cm radial expansion. The nodal CTV was delineated by a 0.5 to 1.5 cm expansion from the nodal GTV. Elective nodal irradiation (ENI) was conducted depending on the primary tumor localization. The planning target volume (PTV) was created with a 0.5 to 1 cm expansion from the CTV.

### 2.3. Data Collection

Tumor staging was conducted using the AJCC 8th edition [14]. Data regarding age, gender, ECOG PS, tumor stage, and inflammatory biomarkers were collected from hospital records. Blood samples were taken within one month before radiotherapy to assess the inflammatory biomarkers.

This study evaluated the C-reactive protein (CRP) levels (mg/L), albumin levels (g/dL), and total bilirubin levels (mg/dL). The neutrophil-to-lymphocyte ratio (NLR), platelet-to-lymphocyte ratio (PLR), and monocyte-to-lymphocyte ratio (MLR) were calculated using the neutrophil count, platelet count, and monocyte count, respectively, divided by the lymphocyte count (all in 10^3^/mL).

The global immune–nutrition–inflammation index (GINI) was calculated according to the formula proposed by Topkan et al. [12], incorporating the CRP, monocyte, platelet, neutrophil, and lymphocyte counts, and albumin levels measured before initiating radiotherapy.
$$GINI=\frac{\left(CRP\times M\times P\times N\right)}{\left(Albumin\times L\right)}$$

The albumin–bilirubin (ALBI) grade was determined based on the ALBI score formula described by Johnson et al. [15].

ALBI score = (log_10_ bilirubin [µmol/L] × 0.66) + (albumin [g/L] × −0.0852).

ALBI grades 1, 2, and 3 were stratified as follows, ALBI score:

≤−2.60 (ALBI grade 1),

>−2.60 to ≤−1.39 (ALBI grade 2),

>−1.39 (ALBI grade 3).

### 2.4. Patient Follow-Up and Response Evaluations

Patients were monitored on a quarterly basis for the initial 2 years following the completion of chemoradiotherapy, at a 6-month interval for the subsequent 3 years, and annually thereafter. To evaluate the local recurrence, progression, and distant metastasis status of the patients, abdomen and chest imaging was conducted utilizing contrast-enhanced CT or PET-CT. The treatment response was assessed by comparing the pre- and post-chemoradiotherapy imaging according to the Response Evaluation Criteria in Solid Tumors (RECIST 1.1) [16].

### 2.5. Statistical Analysis

All the statistical analyses were performed utilizing IBM SPSS version 23.0 software (IBM Corp. 2016, Armonk, NY, USA), with the significance level set at *p* < 0.05. Descriptive statistics were employed to characterize the patient demographics. Comparative analysis of the clinicopathologic and treatment characteristics in the patients classified by the GINI and ALBI grades was conducted utilizing the Chi-square test. Receiver operating characteristic (ROC) curve analysis was conducted to establish the cut-off values for the inflammatory biomarkers, with subsequent calculation of the sensitivity, specificity, area under the curve (AUC), and confidence intervals (CIs). Progression-free survival (PFS) was defined as the period from diagnosis to disease progression, death from any cause in the absence of progression, or confirmation of no progression in surviving patients. Overall survival (OS) was defined as the duration from esophageal cancer diagnosis to either death from any cause or last follow-up. Survival curves were constructed using the Kaplan–Meier method, with the differences assessed via the log-rank test. Univariate Cox proportional hazards regression analyses were utilized to estimate the hazard ratios (HRs) for the PFS and OS. Variables exhibiting statistical significance (*p* < 0.05) in the univariate analyses were incorporated into the multivariate analysis. The hazard risk of individual factors was estimated using the HR with a 95% CI.

## 3. Results

Data from 46 patients were analyzed. Comparative analysis of the clinicopathologic and treatment characteristics of the patients stratified by the GINI and ALBI grades is summarized in Table 1. Most of the patients were female (56.5%) and had an advanced clinical T stage (100.0%), good performance status (82.6%), and regional lymph node metastasis (67.4%). The median fraction and total radiation doses were 1.80 Gy (range: 1.8–2.0 Gy) and 50.4 Gy (range: 40–70 Gy), respectively, and they were delivered in a median of 28 fractions (range: 20–35). Each patient received radiotherapy utilizing the intensity-modulated radiotherapy (IMRT) technique. Thirty-nine patients (84.8%) received definitive radiotherapy and seven patients (15.2%) received neoadjuvant radiotherapy. Chemotherapy was administered to all the patients, and the most frequently administered chemotherapy regimen was carboplatin and paclitaxel (65.2%). Curative surgery was conducted on seven patients who had undergone neoadjuvant treatment. Additionally, six patients (13.0%) required percutaneous endoscopic gastrostomy (PEG) placement.

The cut-off values were determined using ROC curve analysis (Table 2). The cut-off value for the NLR was determined as 2.94 with 60.7% sensitivity and 88.9% specificity, for the PLR as 162.1 with 53.6% sensitivity and 94.4% specificity, for the MLR as 0.27 with 71.4% sensitivity and 61.1% specificity, for CRP (mg/L) as 16.25 with 60.7% sensitivity and 61.1% specificity, for albumin (g/dL) as 4.34 with 71.4% sensitivity and 66.7% specificity, for the GINI as 814.7 with 85.7% sensitivity and 55.6% specificity, for the neutrophil count 10^3^ as 5.6 with 35.7% sensitivity and 100% specificity, for the platelet count 103 as 254 with 57.1% sensitivity and 61.1% specificity, for the monocyte count 10^3^ as 0.48 with 57.1% sensitivity and 55.6% specificity, and for the lymphocyte count 10^3^ as 1.4 with 39.3% sensitivity and 94.4% specificity.

The median follow-up time for this study was 19.5 months (range: 2.6–166.3 months). The one-year PFS and OS rates were 62.2%, and 69.0%, respectively (Figure 1). Progression occurred in 20 patients (43.5%). Of the 20 patients with progression, 15 had local recurrence (LR), 8 had distant metastasis (DM), and 3 had both LR and DM. At the last visit, 18 patients (39.1%) were alive, and 28 patients (60.9%) had died.

The univariate analysis found the platelet (*p* = 0.003) and monocyte (*p* = 0.04) counts to be significant predictive factors for PFS. In the multivariate analysis, the platelet count (*p* = 0.005) was the only independent predictor of PFS. The PFS rates were higher in the platelet count ≤254 group than in the platelet count >254 group (*p* = 0.003) (Table 3). The predictive value of the platelet count for PFS is shown in Figure 2.

The median OS duration was 24.1 months (95% CI, 11.9–36.2). The univariate analysis revealed that the neutrophil count (*p* = 0.04), lymphocyte count (*p* = 0.01), NLR (*p* = 0.005), PLR (*p* = 0.004), CRP (*p* = 0.02), ALBI grade (*p* = 0.01) and GINI (*p* = 0.005) were significant predictive factors for OS (Table 4). In addition, the multivariate analysis determined the GINI to be predictive of improved OS, and this was close to being statistically significant (*p* = 0.08). The OS was significantly higher in the GINI ≤ 814.7 group than the GINI > 814.7 group (*p* = 0.005). The median OS for patients with ALBI grade 2 was 10.2 months, compared to 22.15 months for those with ALBI grade 1 (*p* = 0.01). No cases were observed to be ALBI grade 3. The predictive values of the GINI and ALBI grades for OS are shown in Figure 3 and Figure 4, respectively.

## 4. Discussion

The current study was designed to evaluate the predictive accuracy of the GINI and ALBI grades in patients receiving definitive or neoadjuvant radiotherapy for esophageal cancer. Our results revealed that patients with a GINI score greater than 814.7 exhibited significantly poorer median OS compared to those with a GINI score of 814.7 or less (*p* = 0.005). Additionally, patients with an ALBI grade of 2 had markedly reduced median OS compared to those with an ALBI grade of 1 (*p* = 0.01). These findings provide novel evidence that the GINI and ALBI grades can stratify patients into two distinct prognostic groups with significant differences in the OS. This underscores the previously under-recognized predictive significance of immune, nutritional, and inflammatory markers, which could enhance prediction of the prognosis in patients undergoing definitive or neoadjuvant therapy for esophageal cancer.

The ALBI grade has been reported to be an important prognostic indicator in patients with colorectal cancer, advanced gastric cancer, high-grade glioma, and medulloblastoma [5,6,7,8,9,10]. However, the prognostic and predictive value of the ALBI grade has been predominantly studied in the context of HCC. In the study by Wang et al., the ALBI grade was found to provide more accurate predictions of postoperative liver failure and OS in patients undergoing curative liver resection for HCC compared to the Child–Pugh score [17]. Similarly, in another study, the post-transplant HCC recurrence rates were 10.5%, 15.9%, and 68.2% among patients categorized as ALBI grade 1, 2, and 3, respectively (*p* < 0.001) [4]. In the study by Ho et al., it was demonstrated that the ALBI grade, cirrhosis discriminant score (CDS), and Child–Turcotte–Pugh (CTP) class were the three most precise prognostic markers for assessing the hepatic functional reserve in patients with HCC who were undergoing transarterial chemoembolization (TACE). Among these markers, the ALBI grade was identified as the optimal discriminator of survival across various severity grades [18]. In a systematic review and meta-analysis conducted in May 2022, higher pretreatment ALBI grades were associated with reduced OS. Specifically, the median OS was 12.0 months for patients with ALBI grade 3, compared to 33.5 months for those with ALBI grades 1 and 2, among a cohort of 6538 patients with HCC who were undergoing TACE [19].

The predictive value of the ALBI grade in esophageal cancer has been explored in a limited number of studies to date [20,21,22]. Aoyama et al. evaluated the impact of the preoperative ALBI score on the outcomes in patients with esophageal cancer who underwent curative surgery followed by adjuvant chemotherapy [20]. Their analysis of 121 patients revealed that an ALBI score threshold of −2.7 significantly distinguished recurrence-free survival (RFS) and OS, with lower ALBI scores associated with improved RFS and OS. Shinozuka et al. assessed the preoperative ALBI score as a predictive marker in patients with esophageal squamous cell carcinoma (ESCC) following radical esophagectomy [21]. Their analysis of 449 patients identified a modified ALBI (mALBI) score threshold of −3.33, effectively stratifying patients, with lower scores correlating with improved RFS and disease-specific survival. Kitahama et al. evaluated the predictive value of the preoperative ALBI score in patients with ESCC who were undergoing neoadjuvant chemotherapy (NAC) followed by subtotal esophagectomy [22]. Their analysis of 154 patients demonstrated that a higher pre-NAC ALBI score was significantly associated with worse RFS and OS. Additionally, it was also linked to a reduced pathological response to NAC. Despite the research conducted, the predictive value of the ALBI score in esophageal cancer patients undergoing chemoradiotherapy remains unexplored. Our study represents a significant contribution to the literature in this context. In our study, we employed previously established cut-off points for the ALBI grade [15]. These cut-off values were defined as follows: ≤−2.60 for ALBI grade 1, >−2.60 to ≤−1.39 for ALBI grade 2, and >−1.39 for ALBI grade 3. In the present study, the median OS for patients with ALBI grade 2 was 10.2 months, compared to 22.15 months for those with ALBI grade 1 (*p* = 0.01). No cases were observed to be ALBI grade 3. The ALBI grade has been demonstrated to be a significant predictive factor in esophageal cancer patients undergoing chemoradiotherapy.

Recent advancements in the understanding of the interplay between inflammation and cancer have prompted investigations into novel inflammatory biomarkers and indices across a range of malignancies. Fuca et al. reported the pan-immune inflammation value (PIV) as a novel and comprehensive biomarker [23]. The PIV was calculated as previously described: [neutrophil count (10^3^/mm^3^) × platelet count (10^3^/mm^3^) × monocyte count (10^3^/mm^3^)]/lymphocyte count (10^3^/mm^3^). Baba et al. aimed to examine the relationship between the PIV, tumor immunity, and clinical outcomes in 866 patients with esophageal cancer who underwent surgical resection [24]. Blood samples were obtained and analyzed within 7 days before surgery. Patients with higher PIV values exhibited poorer overall survival outcomes, as demonstrated by both the univariate analysis (*p* < 0.001) and multivariate analysis (*p* = 0.035). Topkan et al. also reported the GINI as a novel and comprehensive biomarker [12]. The GINI was calculated using the following formula: GINI = [C-reactive protein (mg/L) × Platelets (10^3^/mL) × Monocytes (10^3^/mL) × Neutrophils (10^3^/mL)]/[Albumin (g/L) × Lymphocytes (10^3^/mL)]. In the study by Topkan et al., which included a cohort of 802 patients with stage IIIC NSCLC who were undergoing chemoradiotherapy, the optimal pre-chemoradiotherapy GINI cut-off was determined to be 1562. Patients with a GINI score of ≥1562 exhibited significantly shorter locoregional progression-free survival (LRPFS), PFS and OS compared to those with a GINI score of <1562. In our study, the optimal GINI cut-off was found to be 814.7 (AUC: 69%; sensitivity: 85.7%; specificity: 55.6%). Patients with a GINI score > 814.7 demonstrated significantly poorer OS outcomes (*p* = 0.005; HR: 4.73; 95% CI: 1.60–13.9). In the multivariate analysis, this association remained notable though not statistically significant (*p* = 0.08; HR: 3.22; 95% CI: 0.83–12.4). The absence of statistical significance in the multivariate analysis could be ascribed to the limited size of our study cohort.

Our study encountered certain limitations. Firstly, the potential for unintended biases inherent to single-institutional retrospective studies may have influenced our results. Secondly, the absence of an internal validation group may have limited the capacity to accurately assess the predictive significance of the GINI and ALBI scores. Hence, future research endeavors could benefit from incorporating validation cohorts to elucidate these matters within these patient populations; alternatively, subsequent studies should aim to externally validate our findings.

## 5. Conclusions

The findings of the current study suggest that both the GINI and ALBI grades demonstrate promising predictive value in patients with esophageal cancer who are undergoing chemoradiotherapy. Prospective, multi-institutional, and larger cohort studies are required to validate the role of the GINI and ALBI grades in the management of patients with esophageal cancer. Consequently, their application may facilitate the tailored selection of oncologic therapies, thereby optimizing the treatment strategies in the era of personalized oncology.

## Figures and Tables

**Figure 1 curroncol-31-00504-f001:**
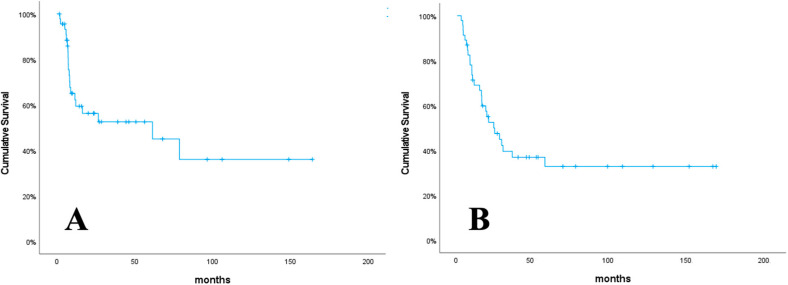
Progression-free survival (**A**), and overall survival (**B**).

**Figure 2 curroncol-31-00504-f002:**
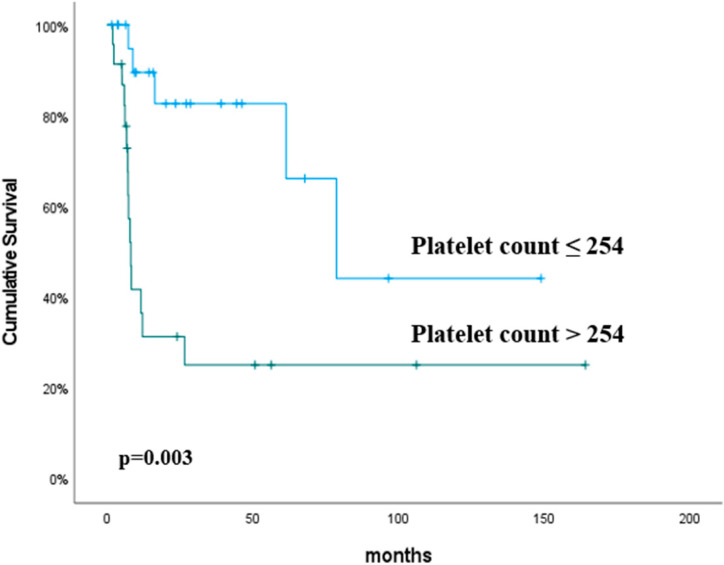
Progression-free survival according to the platelet counts. The *p* values were calculated using the log-rank test.

**Figure 3 curroncol-31-00504-f003:**
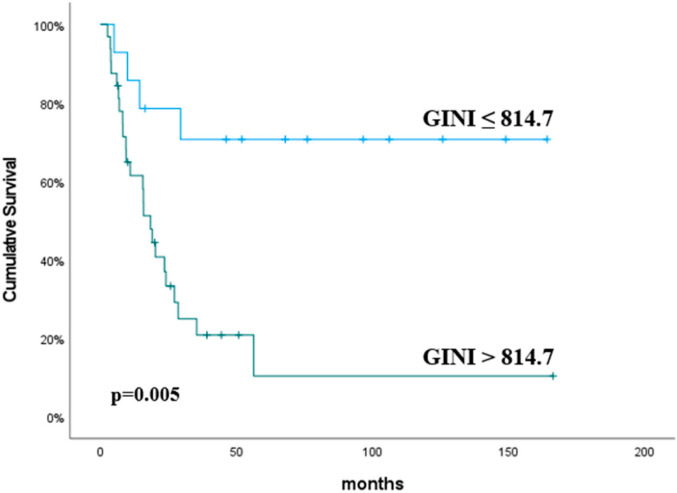
Overall survival according to the GINI. The *p* values were calculated using the log-rank test.

**Figure 4 curroncol-31-00504-f004:**
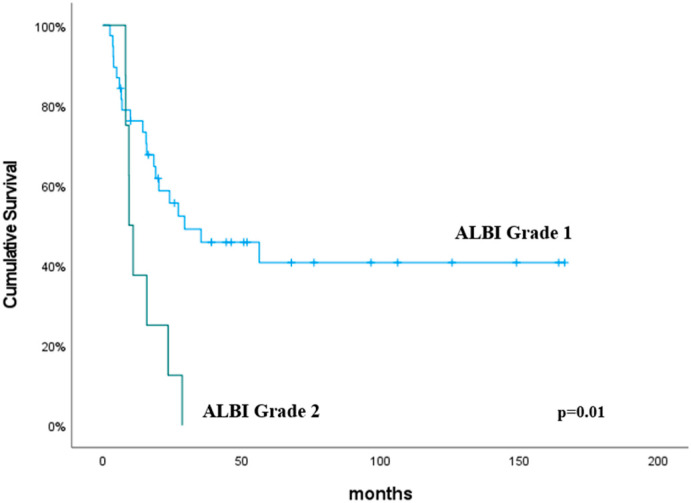
Overall survival according to the ALBI grade. The *p* values were calculated using the log-rank test.

**Table 1 curroncol-31-00504-t001:** Comparative analysis of the clinicopathologic and treatment characteristics of the patients stratified by the GINI and ALBI grades (all patients, *n* = 46).

Variables	GINI			ALBI		
	Low (≤814.7)	High (>814.7)	*p*	Grade 1	Grade 2	*p*
Age						
≤60 years	6 (42.9%)	15 (46.9%)	0.80	18 (47.4%)	3 (37.5%)	0.61
>60 years	8 (57.1%)	17 (53.1%)		20 (52.6%)	5 (62.5%)	
Gender						
Female	3 (21.4%)	17 (53.1%)	0.04	15 (39.5%)	5 (62.5%)	0.23
Male	11 (78.6%)	15 (46.9%)		23 (60.5%)	3 (37.5%)	
ECOG-PS						
0–1	11 (78.6%)	27 (84.4%)	0.01	35 (92.1%)	3 (37.5%)	<0.001
2	3 (21.4%)	5 (15.6%)		3 (7.9%)	5 (62.5%)	
Histology						
SCC	14 (100%)	26 (81.3%)	0.08	34 (89.5%)	6 (75.0%)	0.26
AC	0 (0%)	6 (18.8%)		4 (10.5%)	2 (25.0%)	
T Stage						
3	7 (50.0%)	19 (59.4%)	0.55	20 (52.6%)	6 (75.0%)	0.24
4	7 (50.0%)	13 (40.6%)		18 (47.4%)	2 (25.0%)	
N Stage						
N0	3 (21.4%)	12 (37.5%)	0.28	13 (34.2%)	2 (25.0%)	0.61
N+	11 (78.6%)	20 (62.5%)		25 (65.8%)	6 (75.0%)	
Radiotherapy						
Definitive	13 (92.9%)	26 (81.3%)	0.31	33 (86.8%)	6 (75.0%)	0.39
Neoadjuvant	1 (7.1%)	6 (18.8%)		5 (13.2%)	2 (25.0%)	

GINI, global immune–nutrition–inflammation index; ALBI, albumin–bilirubin grade; ECOG-PS, Eastern Cooperative Oncology Group—Performance Status; SCC, squamous cell carcinoma; AC, adenocarcinoma; N, lymphatic metastasis.

**Table 2 curroncol-31-00504-t002:** ROC curve analysis for the prediction of overall survival.

Variables	Cut-Off	Sensitivity (%)	Specificity (%)	AUC (95% Cl)	*p*
Neutrophil count, 10^3^	5.60	35.7	100.0	0.64 (0.48–0.80)	0.100
Platelet count, 10^3^	254	57.1	61.1	0.54 (0.37–0.71)	0.613
Monocyte count, 10^3^	0.48	57.1	55.6	0.49 (0.32–0.66)	0.964
Lymphocyte count, 10^3^	1.40	39.3	94.4	0.64 (0.48–0.80)	0.100
NLR	2.94	60.7	88.9	0.74 (0.60–0.88)	0.006
PLR	162.1	53.6	94.4	0.71 (0.56–0.86)	0.017
MLR	0.27	71.4	61.1	0.65 (0.49–0.81)	0.075
CRP (mg/L)	16.25	60.7	61.1	0.62 (0.45–0.78)	0.163
Albumin (g/dL)	4.34	71.4	66.7	0.71 (0.56–0.86)	0.014
GINI	814.7	85.7	55.6	0.69 (0.52–0.85)	0.031

CI, confidence interval; NLR, neutrophil–lymphocyte ratio; PLR, platelet–lymphocyte ratio; MLR, monocyte–lymphocyte ratio; CRP, C-reactive protein; GINI, global immune–nutrition–inflammation index.

**Table 3 curroncol-31-00504-t003:** Univariate and multivariate Cox regression analyses for the prediction of progression-free survival.

		Univariate Analysis		Multivariate Analysis	
Variables	Cut-Off	HR (%95 Cl)	*p*	HR (%95 Cl)	*p*
Age (years)	≤60 vs. >60	1.07 (0.44–2.60)	0.86		
Gender	Male vs. Female	1.02 (0.42–2.47)	0.95		
ECOG-PS	0–1 vs. 2	0.88 (0.26–3.03)	0.85		
Histology	SCC vs. AC	2.25 (0.73–6.87)	0.15		
RT	Definitive vs. Neoadj.	0.29 (0.04–2.17)	0.22		
T stage	T3 vs. T4	0.86 (0.35–2.11)	0.75		
N stage	N0 vs. N+	0.93 (0.37–2.34)	0.88		
Neutrophil	≤5.60 vs. >5.60	1.47 (0.52–4.14)	0.46		
Platelet	≤254 vs. >254	4.71 (1.69–13.1)	0.003	4.32 (1.54–12.1)	0.005
Monocyte	≤0.48 vs. >0.48	2.61 (1.04–6.54)	0.04	2.10 (0.79–5.57)	0.13
Lymphocyte	≤1.40 vs. >1.40	1.45 (0.47–4.43)	0.51		
NLR	≤2.94 vs. >2.94	1.00 (0.38–2.58)	1.00		
PLR	≤162.1 vs. >162.1	1.32 (0.51–3.45)	0.55		
MLR	≤0.27 vs. >0.27	1.15 (0.45–2.88)	0.76		
CRP	≤16.25 vs. >16.25	2.19 (0.87–5.50)	0.09		
Albumin	≤4.34 vs. >4.34	0.73 (0.30–1.80)	0.50		
ALBI grade	1 vs. 2	1.61 (0.52–4.94)	0.39		
GINI	≤814.7 vs. >814.7	2.17 (0.74–6.40)	0.15		

HR, hazard ratio; CI, confidence interval; ECOG-PS, Eastern Cooperative Oncology Group—Performance Status; SCC, squamous cell carcinoma; AC, adenocarcinoma; Neoadj, neoadjuvant; N, lymphatic metastasis; NLR, neutrophil–lymphocyte ratio; PLR, platelet–lymphocyte ratio; MLR, monocyte–lymphocyte ratio; CRP, C-reactive protein; ALBI, albumin–bilirubin grade; GINI, global immune–nutrition–inflammation index.

**Table 4 curroncol-31-00504-t004:** Univariate and multivariate Cox regression analyses for the prediction of overall survival.

		Univariate Analysis		Multivariate Analysis	
Variables	Cut-Off	HR (%95 Cl)	*p*	HR (%95 Cl)	*p*
Age (years)	≤60 vs. >60	1.33 (0.62–2.81)	0.45		
Gender	Male vs. Female	0.60 (0.28–1.26)	0.18		
ECOG-PS	0–1 vs. 2	1.77 (0.74–4.18)	0.19		
Histology	SCC vs. AC	1.46 (0.50–4.24)	0.48		
RT	Definitive vs. Neoadj.	1.71 (0.69–4.23)	0.24		
T stage	T3 vs. T4	0.57 (0.26–1.28)	0.17		
N stage	N0 vs. N+	0.72 (0.34–1.55)	0.41		
Neutrophil	≤5.60 vs. >5.60	2.23 (1.02–4.86)	0.04	0.96 (0.35–2.68)	0.95
Platelet	≤254 vs. >254	1.51 (0.71–3.20)	0.27		
Monocyte	≤0.48 vs. >0.48	1.40 (0.66–2.96)	0.37		
Lymphocyte	≤1.40 vs. >1.40	0.37 (0.16–0.84)	0.01	0.86 (0.23–3.17)	0.82
NLR	≤2.94 vs. >2.94	2.99 (1.38–6.46)	0.005	1.91 (0.60–6.08)	0.27
PLR	≤162.1 vs. >162.1	3.09 (1.44–6.66)	0.004	1.14 (0.28–4.60)	0.85
MLR	≤0.27 vs. >0.27	1.82 (0.79–4.15)	0.15		
CRP	≤16.25 vs. >16.25	2.39 (1.10–5.20)	0.02	1.14 (0.42–3.06)	0.79
Albumin	≤4.34 vs. >4.34	0.48 (0.21–1.10)	0.08		
ALBI grade	1 vs. 2	3.07 (1.29–7.25)	0.01	1.28 (0.35–4.59)	0.70
GINI	≤814.7 vs. >814.7	4.73 (1.60–13.9)	0.005	3.22 (0.83–12.4)	0.08

HR, hazard ratio; CI, confidence interval; ECOG-PS, Eastern Cooperative Oncology Group—Performance Status; SCC, squamous cell carcinoma; AC, adenocarcinoma; Neoadj, neoadjuvant; N, lymphatic metastasis; NLR, neutrophil–lymphocyte ratio; PLR, platelet–lymphocyte ratio; MLR, monocyte–lymphocyte ratio; CRP, C-reactive protein; ALBI, albumin–bilirubin grade; GINI, global immune–nutrition–inflammation index.

## Data Availability

The data that support the findings of this study are available from the corresponding author upon reasonable request.

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
