# Peer review of "Predictive Value of GINI and ALBI Grades in Esophageal Cancer Receiving Chemoradiotherapy"

_curroncol, 2024, doi:10.3390/curroncol31110504_

Round 1

Reviewer 1 Report

Comments and Suggestions for Authors

The article entitled “Predictive Value of GINI and ALBI Grade in Esophageal Cancer Receiving Chemoradiotherapy” is an interesting study assessing the predictive efficacy of GINI and ALBI scores among patients receiving chemoradiotherapy for esophageal cancer.

It’s a retrospective study on a small cohort of patients (46) at a single institution who had gone for treatment between Jan 2013 – Jan 2024. Blood samples were used from patients prior to radiotherapy. Different biomarkers were measured for the predictive significance of PFS and OS using both uni- and multi-variate Cox regression analyses.

The GINI and ALBI grades were found to be significantly and independently associated with the OS rates of these patients.

The article is well-conceptualized, comprehensive, and well-written. The authors have explained everything to the point. The outcome of the findings has the potential for clinical relevance but a multi-institutional, multi-national, and larger cohort study should be done. A prospective study including more parameters and different treatment settings will establish these findings for the predictive value of GINI and ALBI grade to be used for association with OS and PFS.

The manuscript in its current form is suitable to be published in the journal.

Author Response

Reviewer Comment:

The article entitled “Predictive Value of GINI and ALBI Grade in Esophageal Cancer Receiving Chemoradiotherapy” is an interesting study assessing the predictive efficacy of GINI and ALBI scores among patients receiving chemoradiotherapy for esophageal cancer.

It’s a retrospective study on a small cohort of patients (46) at a single institution who had gone for treatment between Jan 2013 – Jan 2024. Blood samples were used from patients prior to radiotherapy. Different biomarkers were measured for the predictive significance of PFS and OS using both uni- and multi-variate Cox regression analyses.

The GINI and ALBI grades were found to be significantly and independently associated with the OS rates of these patients.

The article is well-conceptualized, comprehensive, and well-written. The authors have explained everything to the point. The outcome of the findings has the potential for clinical relevance but a multi-institutional, multi-national, and larger cohort study should be done. A prospective study including more parameters and different treatment settings will establish these findings for the predictive value of GINI and ALBI grade to be used for association with OS and PFS.

The manuscript in its current form is suitable to be published in the journal.

Author Response to Reviewers: 

Thank you very much for your constructive and insightful comment and suggestion.

We have acknowledged the single-center, retrospective nature of our study as a limitation and highlighted the need for future validation of our findings through prospective, multicenter studies with larger cohorts.

Reviewer 2 Report

Comments and Suggestions for Authors

In this study, the authors evaluated the predictive value of GINI and ALBI score for esophageal cancer patients undergoing chemoradiotherapy treatment. The study employed univariate and multivariate Cox regression analysis to calculate significance for predicting progression-free survival and overall survival of their patient cohort. The paper found that, in agreement with previous studies utilizing GINI and ALBI in various cancer types and treatment regimens, these metrics successfully categorized patients into subgroups with significantly different survival. 

The manuscript could be further improved by addressing the following:

1. How does PFS, OS, GINI, and ALBI look like for various clinical metrics in the cohort, such as gender, squamous vs adenocarcinoma, and radiotherapy indication? 

2. Please revise Fig 4, which seem to have identical graph repeated several times. 

Author Response

Reviewer Comment:

In this study, the authors evaluated the predictive value of GINI and ALBI score for esophageal cancer patients undergoing chemoradiotherapy treatment. The study employed univariate and multivariate Cox regression analysis to calculate significance for predicting progression-free survival and overall survival of their patient cohort. The paper found that, in agreement with previous studies utilizing GINI and ALBI in various cancer types and treatment regimens, these metrics successfully categorized patients into subgroups with significantly different survival. 

The manuscript could be further improved by addressing the following:

1. How does PFS, OS, GINI, and ALBI look like for various clinical metrics in the cohort, such as gender, squamous vs adenocarcinoma, and radiotherapy indication? 

2. Please revise Fig 4, which seem to have identical graph repeated several times. 

Author Response to Reviewer:

Thank you very much for your constructive and insightful comment and suggestion.

We have conducted a comparative analysis of clinicopathologic and treatment characteristics in patients stratified by GINI and ALBI grades, with the results presented in Table 1.

We have also analyzed the impact of factors such as gender, histopathology, and radiotherapy indications on progression-free survival (PFS) and overall survival (OS), with the results presented in Tables 3 and 4.

Additionally, we have revised Figure 4 and included it once in the manuscript

Reviewer 3 Report

Comments and Suggestions for Authors

The study “Predictive Value of GINI and ALBI Grade in Esophageal Cancer Receiving Chemoradiotherapy” have aimed to assess the predictive efficacy of the Global Immune-Nutrition-Inflammation Index (GINI) and the Albumin-Bilirubin (ALBI) score among patients receiving chemoradiotherapy for esophageal cancer. GINI and/or ALBI has been predictive biomarker for various cancers, this study will be adding knowledge and patient cohort to validate the predictive role in Esophageal Cancer. The study is a single centered study which need to be validated further with a large multi-centered study which has been duly accepted and added as limitation by the authors.

The study is well-conceptualized and conducted. The manuscript is well-written where the available research is clearly presented, discussed, and the conclusion is supported by the evidence presented.

Only suggestion for improvement of the study is that if more information can be added through analysis of pre-existing datasets, it would further validate the conclusion.

Author Response

Reviewer Comment: 

The study “Predictive Value of GINI and ALBI Grade in Esophageal Cancer Receiving Chemoradiotherapy” have aimed to assess the predictive efficacy of the Global Immune-Nutrition-Inflammation Index (GINI) and the Albumin-Bilirubin (ALBI) score among patients receiving chemoradiotherapy for esophageal cancer. GINI and/or ALBI has been predictive biomarker for various cancers, this study will be adding knowledge and patient cohort to validate the predictive role in Esophageal Cancer. The study is a single centered study which need to be validated further with a large multi-centered study which has been duly accepted and added as limitation by the authors.

The study is well-conceptualized and conducted. The manuscript is well-written where the available research is clearly presented, discussed, and the conclusion is supported by the evidence presented.

Only suggestion for improvement of the study is that if more information can be added through analysis of pre-existing datasets, it would further validate the conclusion.

Author Response to Reviewer:

Thank you very much for your constructive and insightful comment and suggestion.

We have acknowledged the single-center, retrospective nature of our study as a limitation and highlighted the need for future validation of our findings through prospective, multicenter studies with larger cohorts.

Additionally, we have conducted a comparative analysis of clinicopathologic and treatment characteristics in patients stratified by GINI and ALBI grades, with the results presented in Table 1.

We have also analyzed the impact of factors such as gender, histopathology, and radiotherapy indications on progression-free survival (PFS) and overall survival (OS), with the results presented in Tables 3 and 4.